# Self-Attentive Moving Average for Time Series Prediction

**Yaxi Su [1], Chaoran Cui [1,\*] and Hao Qu [2]**

[1]  School of Computer Science and Technology, Shandong University of Finance and Economics, No. 7366, East Second Ring Road, Yaojia Sub-District, Jinan 250014, China; suyaxi0301@163.com

[2]  School of Software, Shandong University, Shunhua Road, Jinan 250101, China; quhao_mla@163.com

\*  Correspondence: crcui@sdufe.edu.cn; Tel.: +86-18560132126

**Abstract:** Time series prediction has been studied for decades due to its potential in a wide range of applications. As one of the most popular technical indicators, moving average summarizes the overall changing patterns over a past period and is frequently used to predict the future trend of time series. However, traditional moving average indicators are calculated by averaging the time series data with equal or predefined weights, and ignore the subtle difference in the importance of different time steps. Moreover, unchanged data weights will be applied across different time series, regardless of the differences in their inherent characteristics. In addition, the interaction between different dimensions of different indicators is ignored when using the moving averages of different scales to predict future trends. In this paper, we propose a learning-based moving average indicator, called the self-attentive moving average (*SAMA*). After encoding the input signals of time series based on recurrent neural networks, we introduce the self-attention mechanism to adaptively determine the data weights at different time steps for calculating the moving average. Furthermore, we use multiple self-attention heads to model the *SAMA* indicators of different scales, and finally combine them through a bilinear fusion network for time series prediction. Extensive experiments on two real-world datasets demonstrate the effectiveness of our approach. The data and codes of our work have been released.

**Keywords:** time series prediction; self-attention mechanism; moving average; multi-scale indicator bilinear fusion

## 1. Introduction

Time series analysis has been applied in a wide range of practical problems such as financial market prediction [1], electric utility load forecasting [2], as well as weather and environmental state prediction [3–6]. As it is rather difficult to estimate the exact values of time series, recent studies have mostly paid attention to judging the trend of time series in the future. Typically, time series prediction can be cast as a classification problem in which the goal is to predict the future movement direction of time series, e.g., the rising, falling, and steady trends.

As one of the most popular technical indicators in time series analysis, moving average indicators can summarize the overall changing patterns of time series over a past period in a simple and quick way [7]. Although widely used in different applications, traditional moving average indicators are calculated by averaging the data from different time steps with equal or predefined weights. However, the equal weighting scheme cannot reflect the difference in the importance of different time steps, while manually determining the weights requires a considerable amount of domain knowledge and engineering skills. In addition, unchanged data weights will be applied across different time series, regardless of the differences in their inherent characteristics.

Generally, we can perform time series prediction by comparing the moving averages at different scales. For example, in the stock market, the short-term moving average crosses upward or falls below the long-term moving average, which are usually regarded

as rising and falling signals, respectively, [8]. However, such strategies are still heuristic rules, ignoring the influence of the interaction between the moving average dimensions of different scales, and how to make better use of multi-scale moving averages for time series prediction remains an open question.

To address the above problems, in this paper, we propose a learning-based moving average indicator, called the self-attentive moving average ($SAMA$). Specifically, after encoding the input signals of time series based on recurrent neural networks (RNNs), we introduce the self-attention mechanism [9] to adaptively determine the data weight at each time step for calculating moving averages. In addition, we use multiple self-attention heads to model the moving average indicators of different scales. The bilinear model considers the interaction between the moving average dimensions of different scales and provides a richer representation than the linear model. Numerous studies [10–12] have proved its effectiveness in different fields, and we designed a bilinear fusion sub-network to integrate them for effective time series prediction in an end-to-end manner. Extensive experiments on two real-world datasets validate the rationality of combining multi-scale $SAMA$ indicators, and show that our method significantly outperforms traditional moving average indicators as well as the modern sequential modeling methods for time series prediction. The data and codes of our work are accessible at https://github.com/YY-Susan/SAMA (accessed on 28 February 2022).

In summary, the main contributions of our work are:

- We present a novel learning-based moving average indicator that introduces the self-attention mechanism to adaptively determine the data weight at each time step.
- We use multiple self-attention heads to model the moving average indicators of different scales and use bilinear models to effectively combine them for time series prediction in an end-to-end manner.
- We conduct the experimental evaluation on two real-world datasets and the results demonstrate the effectiveness of our approach.

The remainder of the paper is organized as follows. Section 2 reviews the related work. Section 3 introduces some methodological background regarding moving average indicators. Section 4 details the proposed framework for time series prediction. Experimental setups are described in Section 5. Section 6 concludes our work.

## 2. Related Work

Previously, statistical models such as autoregressive model (AR) [13], moving average model (MA) [14], and autoregressive moving average [15] and its variant ARIMA [16] have been widely used for time series prediction. However, these models cannot describe the nonlinear changes in time series. In order to solve this problem, researchers have resorted to some nonlinear models such as the kernel method [17], Gaussian process [18] and hidden Markov model [19], support vector machines [20], and least squares support vector machine [21] which have the ability to adapt to complex time series. Experiments show that they achieve good efficiency and prediction accuracy in short-term prediction.

With the rapid development of deep learning, CNN [22], RNN [23,24] and its variants, including long short-term memory (LSTM) [4] and gate recurrent unit (GRU) [25], have attracted significant attention and become popular methods for time series modeling. For example, Qin et al. [26] proposed a dual-stage attention-based recurrent neural network (DA-RNN), which contains an encoder and a decoder. The former with an input attention mechanism adaptively extracts the input features of each time step, and the latter with a temporal attention mechanism selects the relevant hidden states of the encoder among all the time steps in different stages. In recent years, the transformer architecture [27] has achieved great success in time series prediction. It completely relies on the self-attention mechanism for time series modeling, and has been reported to achieve promising results.

The self-attention mechanism has been widely used in time series prediction. For example, the continuous failure of high-voltage transmission lines increases the instability of the lines and presents various degrees of hidden security risks, which increase the load of the lines

and the cost of repairing the lines. In response to the above problems, Fahim et al. [28] proposed a self-attention convolutional neural network model (SAT-CNN) which is based on time series imaging for feature extraction and combined self-attention mechanism with CNN to make the model more accurately identify specific types of failures for precise classification. Finally, the effectiveness of the proposed SAT-CNN model is tested with the number of combined input voltage, current, and voltage–current signals at different sampling frequencies. Jin et al. [29] applied the self-attention mechanism to the agricultural field and proposed a bidirectional self-attention encoder–decoder framework (BEDA). Firstly, the wavelet threshold filter and preprocessing are used to denoise the time series, and then the bidirectional long short-term memory network is used to extract the features of the time series. Then the multi-attentional mechanism is introduced into the encoder–decoder framework. Finally, the indoor environmental factors (temperature, humidity, and $CO_2$) are accurately predicted to provide good conditions for crop planting and growth. Experiments show that the framework has good robustness and generalization ability. With the development of modern intelligent transportation systems (ITSs), effectively obtaining the potential spatial pattern and time dynamic traffic flow prediction has become an urgent problem to be solved. Kang et al. [30] proposed a novel spatial-temporal graph self-attentive model (STGSA). The model learns the spatial embedding of the graph through a graph self-attention layer with Gumbel-Softmax technique, and obtains the temporal embedding using an RNN combined with a gated recurrent unit. The effectiveness of the method was demonstrated by experiments on the traffic flow dataset in Langfang, China, in 2014. Wu et al. [31] carried out an in-depth study on the time series prediction of adversarial attack and proposed an adversarial sample generation algorithm based on perturbation. Specifically, the performance of a time series prediction model can be reduced by adding malicious disturbance to time series. Experiments with several time series prediction models on real-world datasets have shown that the proposed method not only deepens the researchers' understanding of time series anti-attack, but also greatly improves the robustness of time series prediction technology. Zheng et al. [32] proposed a temporal change information learning method, in which the mean absolute error (MAE) and mean squared error (MSE) losses are contained in the objective function and use the second-order difference technology in the correlation terms of the objective function. As such, different amplitude errors can be evaluated, and the effects of mutation information and slow change information on the time series can be adaptively obtained. The historical and current moment estimation information is adaptively memorized without introducing redundant hyperparameters. Ding et al. [33] conducted an in-depth study on the fluctuation of time series and proposed a spatial attention fuzzy cognitive map with high-order structure. Firstly, the extended polar fuzzy information granules are used to transform time series into granular sequences with interpretable fluctuation characteristics and then fuzzy cognitive maps are constructed. The attention mechanism is then introduced to make full use of the spatial features and obtain key fluctuation patterns. Finally, a higher-order structure is added to capture the temporal information in the pattern sequence. The effectiveness of the proposed method is verified by a large number of experiments in financial time series.

On the other hand, moving average indicators are important tools for time series analysis and widely used in many research fields, such as natural gas price prediction [34], oil price [35] and foreign exchange prediction [36]. To resolve the deficiencies of traditional moving average indicators, Seng Hansun et al. [37] proposed a new double exponential smoothing method called H-WEMA. Brown's weighted exponential moving average (B-WEMA) [38] has been successfully applied to forex data transaction prediction. In [39], the authors modified and combined the weighting factors of WMA and EMA to form a new weighting scheme for time series prediction. Nakano et al. [40] proposed a stochastic volatility model based on EMA to predict asset returns. The experiment proves that a simple investment strategy with the method is superior to that based on standard EMA. Table 1 briefly summarizes related papers.

**Table 1.** A brief summary of related work.

| Method | Related Work |
|---|---|
| Statistical model | [13–16,37–40] |
| Traditional machine learning model | [17–21] |
| Deep RNN model | [4,23–26,31–33] |
| Deep self-attention model | [27–30] |

## 3. Methodological Background

In order to better express our problem, we declare some symbols in advance. In particular, we use bold capital letters (e.g., $\boldsymbol{X}$), bold lowercase letters (e.g., $\boldsymbol{x}$), lowercase (e.g., $x$), and Greek letters (e.g., $\alpha$) to denote matrices, vectors, scalar, and model hyperparameters, respectively. Denote by $\boldsymbol{X} = [\boldsymbol{x}_1, \boldsymbol{x}_2, \boldsymbol{x}_3, \cdots, \boldsymbol{x}_u]^\top \in \mathbb{R}^{u \times d}$ an input time series, where $u$ is the length of the time series, and $\boldsymbol{x}_t$ is a $d$-dimensional feature vector at the $t$-th time step. In this paper, the objective was to forecast the movement direction of time series at the next time step $u + 1$.

In time series analysis, moving average indicators are one of the most favored tools to measure the overall changing patterns of time series. Typically, there are three traditional moving average indicators frequently used in real scenarios, including simple moving average (SMA) [41], exponential moving average (EMA) [42], and weighted moving average (WMA) [43].

The SMA indicator calculates an average of time series data during a past period. Specifically, the SMA indicator at the $t$-th time step with a lookback window size of $l$ can be calculated as

$$\text{SMA}_l^t = \frac{\sum_{i=1}^l p_{t-i+1}}{l}, \tag{1}$$

where $p_i$ is the data of the $i$-th time step. Intuitively, the SMA indicator is an effective means to eliminate the strong fluctuations of time series. However, it equally treats the data at each time step and ignores the subtle differences of their importance.

As a step further, the EMA indicator adopts an exponential decline manner to weigh the time series data. Particularly, it gives greater weights to recent data than past ones, and can be recursively computed as

$$\text{EMA}_l^t = \alpha \times (p_t - EMA_l^{t-1}) + EMA_l^{t-1}, \tag{2}$$

where $\alpha = \frac{2}{t+1} \in (0, 1)$ represents a degree of weight reduction.

The WMA indicator works in a similar way to EMA but uses the predefined data weights, meaning that:

$$\text{WMA}_l^t = \frac{\sum_{i=1}^l p_{t-i+1} w_i}{\sum_{i=1}^l w_i}, \tag{3}$$

where $w_i$ denotes the predefined weight. For notational simplicity, we shall omit the superscript $t$ in the following.

In time series analysis, it is widely believed that the intersection of moving average indicators at different scales suggests a certain trend signal [44]. For example, when a short-term moving average crosses a long-term moving average from bottom to top, it means that an upward signal of a time series can be generated. On the contrary, if a short-term moving average crosses a long-term moving average from top to bottom, a downward time series trend may be predicted. However, such a strategy is mainly formed on humans' subjective experience, and may not be applicable to different time series.

## 4. Method

In this paper, we propose an end-to-end method. Firstly, we demonstrate the process of dynamically modeling time series and details in self-attentive moving average ($SAMA$). Then, different scales' $SAMA$ indicators are integrated through a bilinear fusion sub-network for time series prediction. The overall framework is shown in Figure 1.

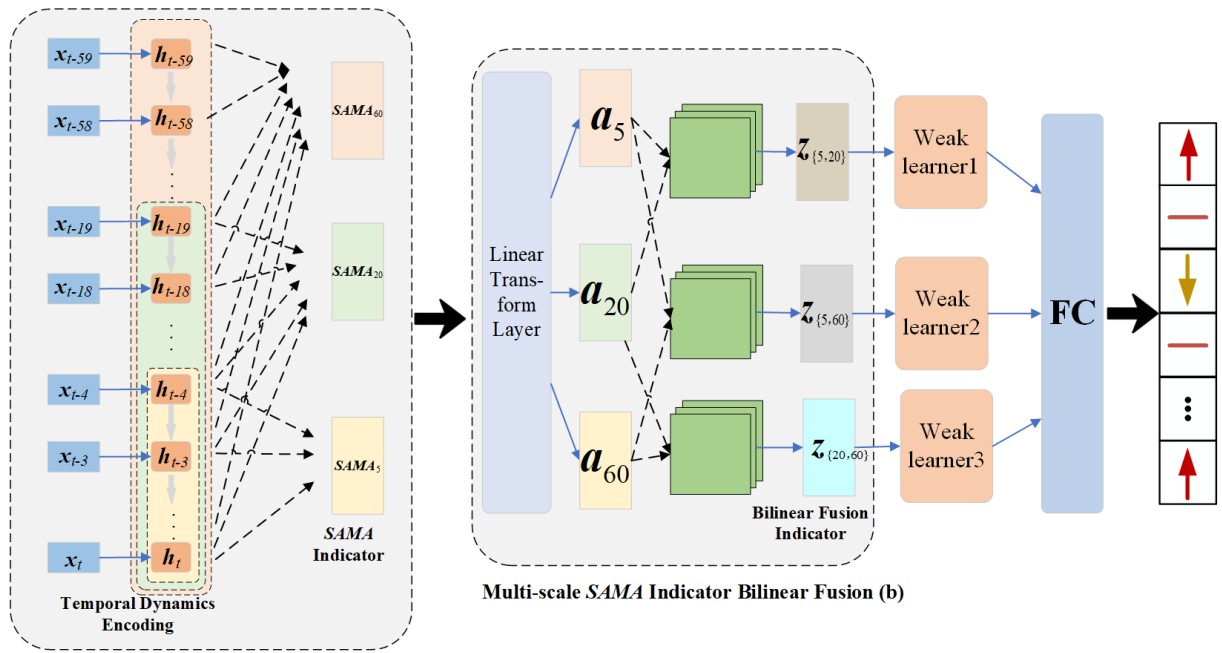

**Figure 1.** The overall framework. In part (**a**), the input signal of a time series is encoded through a recurrent neural network, and then a self-attention mechanism is introduced to adaptively determine the weight of the data at each time step to calculate the $SAMA$ indicators. Multiple self-attention heads are used to model multiple moving average indicators of different scales. In part (**b**), the bilinear fusion method is used to fuse information between the dimensions of any two indicators.

### 4.1. Time Series Encoding

As we all know, the historical data of time series plays a pivotal role in predicting its future trends. In this paper, we use the LSTM [45] model to encode the historical data of time series. Compared with the vanilla RNN, the LSTM model has a long-term memory function, which could alleviate the problems of gradient vanishing and exploding for long-term sequence modeling. At the $t$-th time step, we input the feature vector $x_t$ into the LSTM model. Formally, the LSTM model performs the calculations as follows:

$$
\begin{aligned}
f_t &= \sigma(W_f[h_{t-1}, x_t] + b_f) \\
i_t &= \sigma(W_i[h_{t-1}, x_t] + b_i) \\
o_t &= \sigma(W_o[h_{t-1}, x_t] + b_o) \\
\tilde{c}_t &= \tanh(W_c[h_{t-1}, x_t] + b_c) \\
c_t &= f_t \odot c_{t-1} + i_t \odot \tilde{c}_t \\
h_t &= o_t \odot \tanh(c_t)
\end{aligned}
\tag{4}
$$

Here, $h_{t-1}$ is a hidden state containing all the information up to the $(t-1)$-th time step. $h_{t-1}$ and $x_t$ are concatenated and converted into a forget gate $f_t$, an input gate $i_t$, and an output gate $o_t$, respectively. In addition, $h_{t-1}$ and $x_t$ are also used to generate a candidate cell state $\tilde{c}_t$ which represents the new information to be added. Then, $c_{t-1}$ and $\tilde{c}_t$ are combined to form $c_t$, and $f_t$ and $i_t$ serve as the balance factors in this procedure. Finally, $o_t$ is multiplied by $c_t$ to output the current hidden state $h_t$. In Equation (4), $W_f$, $W_i$,

$W_o$, and $W_c$ are the parameters to be learned, $\odot$ represents the Hadamard product, and $\sigma(\cdot)$ and $\tanh(\cdot)$ are the sigmoid and tanh activation functions, respectively.

### 4.2. Self-Attentive Moving Average

As described above, a core issue in calculating moving average indicators is to choose the appropriate data weights at different time steps. In this paper, we propose a new type of moving average indicator, called self-attentive moving average ($SAMA$), which introduces the self-attention mechanism [9] to adaptively determine the data weights. Specifically, the input signals of time series are encoded through the LSTM model and the hidden state of each time step is obtained. When computing the $SAMA$ indicator at the $t$-th time step, the data weight $w_i$ is measured by examining the extent to which the hidden state $h_{t-i+1}$ is compatible with a query reference. Based on the recent bias hypothesis [46] that the future trend of the time series has a strong correlation with its recent volatility, we select the hidden state $h_t$ of the current time step as the query reference, and $w_i$ is thus defined as

$$w_i = \frac{\exp(s(h_t, h_{t-i+1}))}{\sum_{j=1}^{l} \exp(s(h_t, h_{t-j+1}))}, \tag{5}$$

where:

$$s(h_t, h_{t-i+1}) = \frac{(Qh_t)^\top (Kh_{t-i+1})}{\sqrt{d}} \tag{6}$$

is a compatibility function that transforms $h_t$ and $h_{t-i+1}$ into a latent space with the parameter matrices $Q$ and $K$, and $d$ is the dimension of the latent space.

Finally, $SAMA_l$ can be computed as the weighted sum of the transformed hidden state at each time step, meaning that:

$$SAMA_l = \sum_{i=1}^{l} w_i V h_{t-i+1}, \tag{7}$$

where $V$ is a transformation matrix to be learned.

### 4.3. Multi-Scale SAMA Bilinear Fusion

In this study, we use multiple self-attention heads [9] to compute the $SAMA$ indicators of different scales, and further combine them to forecast the future trend of time series. Specifically, the 5-step, 20-step, and 60-step moving averages are simultaneously modeled, i.e., $SAMA_5$, $SAMA_{20}$, and $SAMA_{60}$ indicators, which capture the past weekly, monthly, and seasonal dynamic patterns of day-frequency time series, respectively.

Generally speaking, we predict the moving direction of time series by comparing different scales' moving average indicators. For example, in the stock market, the short-term moving average crosses upward or falls below the long-term moving average, which are usually regarded as rising and falling signals, respectively. However, such heuristic strategies are highly subjective and do not make full use of the characteristics of different scales' moving average indicators. In this paper, we resort to the bilinear fusion [10] to make the different dimensions of different scales' indicators interact with each other and obtain richer representations. To be specific, we input the three moving averages $SAMA_5$, $SAMA_{20}$, and $SAMA_{60}$ into the linear transformation layer, and perform dimensional expansion operations on them, respectively, to obtain vectors $a_5$, $a_{20}$, and $a_{60}$. Then, we send them to the bilinear fusion layer and the indicators of different scales are aggregated to a matrix through the outer product. As such, the interaction and fusion between each pair of indicators $a_i$ and $a_j$ are carried out, and the bilinear vector $z_{\{i,j\}}$ is finally obtained:

$$z_{\{i,j\}} = flat(a_i \otimes a_j), \tag{8}$$

where $\otimes$ and $flat$ denote the outer product and flattening operation, respectively. $i$, $j \in \{5, 20, 60\}$ and $i \neq j$. Finally, we adopt the idea of ensemble learning to make time series prediction. We input the obtained bilinear fusion vectors into three weak learners to adaptively learn the corresponding weight of the bilinear fusion vectors. Then, we send them to a strong learner with a fully connected layer for future trend prediction:

$$\hat{y} = W_{FC}[W_1 z_{\{5,20\}}, W_2 z_{\{5,60\}}, W_3 z_{\{20,60\}}], \tag{9}$$

where $W_1$, $W_2$, $W_3$, and $W_{FC}$ are the weight matrices to be learned.

We consider three movement directions of time series, namely the rising, falling, and steady trends. We use cross entropy as the loss function to penalize the deviation of the prediction from the ground-truth. The $\hat{y}$ donates the predicted probability distribution over different trends at the next time step, and $y$ is a one-hot vector indicating the true trend label. The cross entropy loss can be computed as

$$l(\hat{y}, y) = \sum_i y_i \log \hat{y}_i, \tag{10}$$

where $i$ is the dimension index. The optimal $SAMA$ indicator can be determined by minimizing the loss over all time series.

## 5. Experiments

### 5.1. Data Collection

In this paper, we conducted experimental evaluations on a publicly available stock dataset [47] and air quality dataset (https://download.csdn.net/download/godspeedch/10627195$?$utm$_$source=iteye$_$new, accessed on 28 February 2022), respectively. For both datasets, we take each day as a time step. Specifically, the stock dataset contains 1026 stocks collected from the NASDAQ market, with trading records consisting of the opening price, closing price, highest price, lowest price, and trading volume of each day between 2 January 2013 and 12 August 2017. The air quality dataset contains the statistics of $PM_{2.5}$, $PM_{10}$, $SO_2$, $CO$, $NO_2$, $O_3$, and AQI per day from 2 December 2013 to 31 October 2018. During data preprocessing, Min-Max normalization is used to normalize time series data to the interval of [0,1]. We divide the data into the training set and testing set in chronological order by the ratio of 4:1.

### 5.2. Evaluation Methodology

A time series prediction problem can be solved as a classification problem or a regression problem. The change pattern of time series is often unstable and irregular, so it is difficult to accurately predict the value at a specific time step. In previous studies [26,48,49], researchers have mainly focused on predicting the future trend of time series, transforming time series prediction into a classification problem to solve. In our study, the trend of the next time step is defined as one of the directions of rising ($+1$), falling ($-1$), and steady ($0$). The ground-truth label is determined based on the change ratio of the closing price and AQI for the stock and air quality datasets, respectively, meaning that:

$$y = \begin{cases} +1, & \text{if } \dfrac{v_{t+1} - v_t}{v_t} \geq \alpha; \\ -1, & \text{if } \dfrac{v_{t+1} - v_t}{v_t} \leq -\alpha; \\ 0, & \text{otherwise}, \end{cases} \tag{11}$$

where $v_t$ and $v_{t+1}$ are the values of closing prices or AQI at the $t$-th and $(t+1)$-th time step, respectively. $\alpha$ is a threshold and set to be 0.50% in line with the previous work [48]. We use the accuracy, precision, recall, and $F_1$ score as metrics to evaluate the performance of different algorithms. To further analyze the results, we performed paired a t-test to compare the difference between our method and the other existing methods, and found that the im-

provement of our method is statistically significant at the significance level of 0.05 [50]. For each method compared in our experiments, we repeat the training and testing procedures five times, and report the average performance to alleviate the fluctuations caused by random initializations.

### 5.3. Baseline

We compared the $SAMA$ indicators with different time series' prediction algorithms. Specifically, we first compare the results of single $SAMA$ indicators and different scales' fusion to prove the effectiveness of multi-scale bilinear fusion:

- $SAMA_l$: the single-scale $SAMA$ indicator is used for time series prediction.
- $SAMA_l | SAMA_{l'}$: any two scales $SAMA$ indicators are simply spliced to make time series prediction.
- $SAMA_5 | SAMA_{20} | SAMA_{60}$: the $SAMA$ indicators of three different scales are spliced to make time series predictions.

We then compare the model with three traditional technical indicators and make further comparison with the depth sequence model:

- SMA [41]: the SMA indicator calculates the average value of the time series data during a period of time.
- EMA [42]: the EMA indicator is calculated based on the principle that the weight of the time series data decreases exponentially.
- WMA [43]: the WMA indicator gives predefined weights to different time series data.
- TCN [51]: the model combines CNN and RNN structure. The input sequence of arbitrary length is modeled by causal convolution, expansion convolution and residual joining.
- DA-RNN [26]: the model is based on a recurrent neural network of dual-stage attention. The encoder with an input attention mechanism and the decoder with time attention are used to adaptively extract the input features and select the encoding-related hidden states at all time steps.
- Transformer [9]: the transformer completely relies on the self-attention mechanism to calculate the input and output representations, and obtains the weight of each value by calculating the similarity between the query and the corresponding key.
- LSTM [52]: LSTM models time series through recurrent neural networks and predicts future trends.

### 5.4. Effectiveness of Multi-Scale SAMA Bilinear Fusion

In our study, we propose combining multiple $SAMA$ indicators of different scales for time series prediction. Tables 2 and 3 report the comparison results between each individual $SAMA$ indicator and the fusion of multi-scale ones in different ways, from which we can make the following observations:

- Overall, combining multi-scale $SAMA$ indicators offers better performance than utilizing one of them alone, leading to, for example, at least 3.19% and 2.85% relative improvements in terms of $F_1$ score on the stock and air quality datasets, respectively. The results suggest the necessity of simultaneously exploiting the $SAMA$ indicators of different scales for enhancing time series prediction. In addition, the performance of time series trend prediction by combining the three indicators $SAMA_5$, $SAMA_{20}$, and $SAMA_{60}$ deteriorates, which may be caused by the following two reasons. The $SAMA_{20}$ indicator is between the $SAMA_5$ and $SAMA_{60}$ indicators, which can reflect the fluctuations of time series in the past month. Since the $SAMA_{20}$ indicator is not sensitive to the short-term fluctuation of the time series and cannot adequately reflect the long-term trend change, some redundant information is introduced when the three indicators are combined for trend prediction, which reduces the accuracy of the prediction. On the other hand, we simply splice the three indicators to obtain a unified representation through the fully connected layer as an excessively simple combination method may lead to performance deterioration.

- It can be seen that the bilinear fusion of $SAMA_5$, $SAMA_{20}$, and $SAMA_{60}$ achieves the best result among all multi-scale competitors. This may be because $SAMA_5$ could reflect the recent short-term fluctuations of time series: on the contrary, $SAMA_{20}$ and $SAMA_{60}$ capture the overall mid-term and long-term dynamics of the past period. In the process of fusion, we interact with the three different indicators and comprehensively consider the dimensions of different scale indicators to better predict the future trend of the time series.

**Table 2.** Performance of multi-scale $SAMA$ indicator fusion (stock dataset).

| Method | Stock Dataset | | | |
| --- | --- | --- | --- | --- |
| | **Accuracy** | **Precision** | **Recall** | $F_1$ |
| $SAMA_5$ | 42.13% | 41.44% | 41.08% | 39.58% |
| $SAMA_{20}$ | 42.63% | 41.98% | 41.56% | 40.10% |
| $SAMA_{60}$ | 42.68% | 41.89% | 41.50% | 39.81% |
| $SAMA_5 \mid SAMA_{20}$ | 42.71% | 42.17% | 41.59% | 39.90% |
| $SAMA_5 \mid SAMA_{60}$ | 42.76% | 42.20% | 41.65% | 40.27% |
| $SAMA_{20} \mid SAMA_{60}$ | 42.71% | 42.00% | 41.63% | 40.39% |
| $SAMA_5 \mid SAMA_{20} \mid SAMA_{60}$ | 42.75% | 42.13% | 41.61% | 40.12% |
| $(SAMA_5, SAMA_{20}, SAMA_{60})_{mix}$ | **42.82%** | **42.41%** | **41.98%** | **41.38%** |

The best result is indicated in bold and the second best result is underlined. This convention is also adopted in the following table. Furthermore, $\mid$ denotes the combination of different scales.

**Table 3.** Performance of multi-scale $SAMA$ indicator fusion (air quality dataset).

| Method | Air Quality Dataset | | | |
| --- | --- | --- | --- | --- |
| | **Accuracy** | **Precision** | **Recall** | $F_1$ |
| $SAMA_5$ | 69.66% | 46.22% | 46.35% | 46.26% |
| $SAMA_{20}$ | 67.86% | 45.03% | 45.39% | 45.20% |
| $SAMA_{60}$ | 69.40% | 45.88% | 45.73% | 45.74% |
| $SAMA_5 \mid SAMA_{20}$ | 68.83% | 45.65% | 45.78% | 45.69% |
| $SAMA_5 \mid SAMA_{60}$ | 70.15% | 46.55% | 46.94% | 46.67% |
| $SAMA_{20} \mid SAMA_{60}$ | 69.78% | 46.19% | 45.81% | 45.86% |
| $SAMA_5 \mid SAMA_{20} \mid SAMA_{60}$ | 69.40% | 45.88% | 46.01% | 45.94% |
| $(SAMA_5, SAMA_{20}, SAMA_{60})_{mix}$ | **71.64%** | **47.45%** | **47.75%** | **47.58%** |

*5.5. Performance Comparison*

We compare our approach with the traditional moving average indicators, i.e., SMA, EMA, and WMA, as well as the modern sequential modeling methods including LSTM and transformer, in terms of time series prediction. Table 4 lists the performance comparison of different methods. From the results, we can see that:

- In most cases, traditional moving average indicators fall considerably behind the deep model-based methods. This highlights the merit of deep learning techniques for time series prediction.
- Among deep sequence models, transformer and our approach are both superior to TCN, DA-RNN, and LSTM. This underlines the benefit of introducing the self-attention mechanism to capture the dynamics of time series. On the other hand, our approach obtains the best results on both datasets. More precisely, it exceeds the transformer by an average of nearly 1.67%, 1.21%, 1.15%, and 1.49% in terms of accuracy, precision, recall, and $F_1$ score, respectively. The results clearly demonstrate the effectiveness of our approach for time series prediction.

**Table 4.** Performance comparison between different methods.

| Method | Stock Dataset | | | | Air Quality Dataset | | | |
|---|---|---|---|---|---|---|---|---|
| | Accuracy | Precision | Recall | $F_1$ | Accuracy | Precision | Recall | $F_1$ |
| SMA | 39.23% | 35.83% | 37.12% | 29.89% | 67.92% | 45.98% | 44.27% | 44.00% |
| EMA | 39.26% | 37.25% | 37.09% | 29.51% | 67.91% | 44.91% | 44.43% | 44.46% |
| WMA | 39.98% | 38.02% | 37.92% | 30.11% | 69.03% | 45.64% | 45.37% | 45.40% |
| TCN | 40.90% | 38.89% | 39.24% | 33.91% | 66.42% | 44.05% | 44.40% | 44.15% |
| DA-RNN | 40.82% | 39.62% | 39.17% | 34.21% | 67.16% | 44.84% | 43.28% | 43.17% |
| LSTM | 41.93% | 40.58% | 40.43% | 37.46% | 68.73% | 45.59% | 45.70% | 45.61% |
| Transformer | 41.97% | 41.53% | 41.12% | 39.89% | 69.16% | 45.92% | 46.32% | 46.10% |
| Ours | **42.82%** | **42.41%** | **41.98%** | **41.38%** | **71.64%** | **47.45%** | **47.75%** | **47.58%** |

## 6. Conclusions

In this paper, we propose a new learning-based moving average indicator $SAMA$. After dynamically encoding the input signals of time series, we introduced the self-attention mechanism to adaptively determine the data weights of each time step for the moving average calculation. We generated the $SAMA$ indicators of different scales with multiple self-attention heads and combined them through a bilinear fusion network for time series prediction in an end-to-end manner. The experimental results on real-world datasets demonstrate the effectiveness of our framework.

In the future, we plan to explore more complicated network architectures to integrate moving average indicators of different scales. Many researchers [53–57] have explored the opportunities and challenges of big data. Boulesteix et al. [58] explained why statistical models should be evaluated using large datasets, thus we will also verify our approach on large-scale datasets and adapt it to other tasks in the field of time series analysis. In addition, the Bert model can be used to analyze the sentiment of text information to obtain sentiment embedding. At the same time, the method proposed in this paper can be used to calculate the SAMA indicators and fuse different scales' indicators to obtain the temporal embedding and finally combine the two to make a time series prediction.

**Author Contributions:** Conceptualization, Y.S. and C.C.; methodology, Y.S.; validation, H.Q.; formal analysis, H.Q.; data curation, Y.S.; writing—original draft preparation, Y.S.; writing—review and editing, C.C. All authors have read and agreed to the published version of the manuscript.

**Funding:** This work was supported by the National Natural Science Foundation of China under Grant 62077033 and Grant 61701281, and by the Fostering Project of Dominant Discipline and Talent Team of Shandong Province Higher Education Institutions.

**Institutional Review Board Statement:** Not applicable.

**Informed Consent Statement:** Not applicable.

**Data Availability Statement:** The datasets and code used in this study are linked in this paper.

**Conflicts of Interest:** The authors declare no conflict of interest.

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
