# Peer review of "Self-Attentive Moving Average for Time Series Prediction"

_applsci, doi:10.3390/app12073602_

Round 1

Reviewer 1 Report

Summary

The paper proposes a learning-based moving average indicator under the name Self-Attentive Moving Average (SAMA) for predicting the future trend of time series. More precisely, the paper proposes the utilization of recurrent neural networks for encoding the input signals of time series, followed by the utilization of a self-attention mechanism to adaptively determine the data weights at different time steps for calculating the moving average, while traditional moving average indicators are calculated by averaging the time series data with equal or predefined weights. Furthermore, the paper proposes the utilization of multiple self-attention heads for modelling SAMA indicators of different scales, and their subsequent combination through a bilinear fusion network for time series prediction. Experiments using two real-world datasets are presented. These datasets and the study’s codes have been released on GitHub.

General evaluation

I find that the paper is interesting, meaningful and well-prepared, in general terms.

I also believe that the release of the study’s data and codes on GitHub should be much appreciated.

I provide a few minor comments that could be addressed for improving the presentation of the research. These comments are provided here below.

Minor comments

  • To my view, the entire “Discussion” section should be carefully revised. In fact, this section currently reads more like a part of the “Introduction” section, as it presents (part of) the literature background of the conducted research. In the “Discussion” section the proposed methods should be discussed compared to previous methods and/or the most important findings of the study should be discussed in light of the existing knowledge in the field.
  • Based on the above, perhaps the current “Discussion” section could be moved to the “Introduction” section. Then, an entirely new “Discussion” section could be formulated.
  • Maybe the title of Section 2 (i.e., “Preliminary”) could be changed to something like “Methodological background”. The current title is bit confusing, to my view.
  • In lines 24−26, it is written that “Typically, time series prediction can be cast as a classification problem, in which the goal is to predict the future movement direction of time series, e.g., the rising, falling, and steady trends”. Please note here that time series prediction can be either a classification or a regression problem.
  • The explanations for bolded and underlined values are missing from Tables 2 and 3.
  • In lines 259−260, it is written that “Besides, we will verify our approach on large-scale datasets, and adapt it to other tasks in the field of time series analysis”. I think that the first part of this sentence could be briefly discussed in light of Boulesteix et al. (2018), who explain why statistical models should be evaluated using large datasets.

Reference

Boulesteix AL, Binder H, Abrahamowicz M, Sauerbrei W (2018) Simulation Panel of the STRATOS Initiative. On the necessity and design of studies comparing statistical methods. Biometrical Journal 60:216–218. doi:10.1002/bimj.201700129

Reviewer 2 Report

The manuscript is like an experimental report. The architecture of the manuscript needs to be improved. It would be better if the authors introduced the methods before the experiments. The expression of the way should be separated from the experiment. Thus, it requires significant revisions.

  • What are the authors' scientific contributions? The hybrid of several DL models is not innovative enough. Many studies have proposed such hybrid DL models. Would you mind clarifying them?
  • After introducing each relevant work in the Introduction section, the authors should add comments from the cited publications. The explicit thinking/consideration of why the proposed technique can obtain more positive findings is what readers expect from a convincing literature study. This is the authors' contribution in its entirety. Furthermore, the authors could provide a more thorough critical literature analysis for energy forecasting to highlight the shortcomings of existing methodologies, describe the mainstream of research direction clearly, and how prior studies performed. Which approaches should you use? Which of the problems has yet to be resolved? Why is the proposed method appropriate for solving the critical problem? More convincing studies in the literature are needed to clearly show the state-of-the-art advancement for energy forecasting from famous international journals such as MDPI, Elsevier, or IEEE.
  • The authors have to briefly summarize the relevant research cases described in the Introduction section using a table to help readers understand.
  • The authors should more effectively introduce their proposed research framework: some crucial brief explanation concerning the text with a total research flowchart or framework diagram for each proposed algorithm to show how these employed models are working to receive the experimental results. It isn't easy to figure out how the recommended solutions work.
  • In Section 4, the authors should employ additional alternative models as benchmarking models and conduct some statistical tests to ensure the superiority of the suggested technique. For tackling regression issues, boosting algorithms (e.g., gradient boosting machine (GBM), extreme gradient boost (XGB), LightGBM, and CatBoost) have demonstrated excellent prediction performance. On the other hand, the authors did not consider these algorithms. The authors could employ more alternative models as benchmarking models; how can the authors ensure that their results are superior to others?
  • The authors can consult the following references for statistical tests, such as the Wilcoxon signed-rank and Freidman tests.
  • It is well-known that all neural networks are based on unknown parameters. How did you calibrate the proposed model to make sure you always used the optimal values for the parameters, such as the time steps in RNNs? Did you use a trial-and-error approach or a more systematic approach? The authors should provide more details and justification in this regard.
  • Please show the prediction models' training and testing times of the prediction models.
  • A complete introduction to the most recent studies after 2020 is required.
  • Figure 1 is only made up of overly simplistic statements. The authors' contribution is minor when only the flowchart in Figure 1 is considered.
  • The authors should motivate the readers by giving a variety of experimental results. It can only be presumed that the contribution of the presented cases is insignificant on its own.
  • The authors must raise the language of the article in general. Proofreading by multiple people familiar with the field of study will help in this regard.
  • I wonder if the proposed method can be applied to other regions with different mechanical systems and occupant profiles. It is challenging to trust the results of this experiment fully. In addition, deep learning analysis requires repeated experiments because the predicted value varies depending on how the initial weight is set.
  • It would be better if the results were more detailed using more graphs. For example, the authors could give error graphs for the monthly forecast, the days of the week forecast, and others.
  • The authors must conduct exploratory data analysis, such as histograms, scatter plots, and box plots, and explanations that are simple enough for the reader to understand, such as concept links and cluster analysis tables.
  • The description of the input variable is unclear. Using a table, the authors should clearly explain the input variables and the output variable. What input variables did the authors consider?
  • Did the authors consider the Min-Max normalization in Section 3? 
  • This problem is complex to tackle given the limited memory and storage of mobile devices, and why only ML is fit for the task. This element requires extra attention in the Discussion and Conclusion sections of the paper, referring to the following: 1) Md Ibrahim Khalil, R. Young Chul Kim, ChaeYun Seo (2020). Challenges and Opportunities of Big Data. JOURNAL OF PLATFORM TECHNOLOGY, 8(2), 3-9;
    2) S.Vimal, Y. Harold Robinson, M.Kaliappan, Subbulakshmi Pasupathi, A.Suresh (2021). Q Learning MDP Approach to Mitigate Jamming Attack Using Stochastic Game Theory Modelling With WQLA in Cognitive Radio Networks. JOURNAL OF PLATFORM TECHNOLOGY, 9(1), 3-14;
    3) S.Vimal, Jesuva Arockiadoss S, Bharathiraja S, Guru S, V.Jackins (2021). REDUCING LATENCY IN SMART MANUFACTURING SERVICE SYSTEM USING EDGE COMPUTING. JOURNAL OF PLATFORM TECHNOLOGY, 9(1), 15-22;
    4) Han, Y., & Hong, B. W. (2021). Deep learning based on fourier convolutional neural network incorporating random kernels. Electronics, 10(16), 2004.
  • Because the authors must improve the conclusions explaining how this research helps expand previous study and body of knowledge, the authors must rewrite the introduction and conclusions specifically as below:
    1) The Introduction did not have a smooth association between paragraphs, and this study's background is unclear.
    2) The Conclusions have been made too simple, and there are no limitations and directions for improvement for this study.

Round 2

Reviewer 2 Report

The author addresses all the questions raised in the first round in the current version of the manuscript. This document is suitable for publication in Applied Sciences.